

# The role of N6-methyladenosine (m$^6$A) RNA methylation modification in kidney diseases: from mechanism to therapeutic potential

Shaowen Guo[1,2], Wenjun Wang[1,2], Gaopan Lv[1,2], Yun Ling[1] and Meifeng Zhu[2]

[1] Nanjing University of Traditional Chinese Medicine, Nanjing, China
[2] Changzhou Hospital of Chinese Medicine Affiliated to Nanjing University of Chinese Medicine, Nanjing University of Traditional Chinese Medicine, Changzhou, China

Corresponding authors
Yun Ling, yunlizzy@njucm.edu.cn
Meifeng Zhu,
zhumeifeng@njucm.edu.cn

## ABSTRACT

**Background:** Kidney disease is a major global health issue, causing numerous deaths and a loss of life years. This prompts us to explore potential targets or mechanisms that may increase the likelihood of diagnosing and treating kidney diseases. N6-methyladenosine (m$^6$A) modifications dynamically regulate RNA through "writer" enzymes, "eraser" enzymes, and "reader" proteins, influencing its processing, stability, and translation efficiency. In cases of kidney disease, there is a likelihood that m$^6$A methylation is a significant contributor to the pathological mechanisms of acute kidney injury (AKI), chronic kidney disease (CKD), diabetic kidney disease (DKD), renal cell carcinoma (RCC), and lupus nephritis (LN). In this article, we explore the role and mechanisms of m$^6$A methylation in kidney diseases and its applications in the treatment of kidney diseases.

**Methodology:** This review systematically evaluated the therapeutic relevance of m$^6$A methylation in renal diseases using a targeted search strategy across multiple databases (Scopus, PubMed, Web of Science, Google Scholar, bioRxiv, medRxiv) from January 1970 to May 2025. Study quality was assessed, and critical data elements were cataloged to ensure rigor.

**Results:** The current research investigates m$^6$A methylation's role in kidney diseases, highlighting its significant impact on regulating gene expression, affecting cell signaling pathways, and modulating inflammation. In AKI, changes in m$^6$A modification levels are closely associated with the severity of kidney damage. Specifically, m$^6$A regulators such as METTL3 and FTO influence the progression of AKI by affecting gene expression, oxidative stress, and inflammation. Regarding CKD, decreased m$^6$A modification levels could potentially cause atypical gene expression in cells, thus impairing normal cellular functions. In diabetic nephropathy (DN), dysregulated expression of genes linked to m$^6$A methylation is closely associated with renal hypertrophy, proteinuria, and glomerulosclerosis. In LN, alterations in m$^6$A regulator expression are strongly linked to glomerular filtration rate (GFR).

**Conclusions:** Emerging studies link dysregulated m$^6$A machinery to diverse kidney diseases, including acute/chronic kidney injury (WTAP/METTL3/FTO in oxidative stress and fibrosis), and diabetic nephropathy (METTL14/FTO polymorphisms in

susceptibility). Mechanistically, m$^6$A modulates TGF-β signaling, inflammatory responses, and gene networks underlying disease progression. Despite therapeutic promise, challenges persist in methodological standardization and understanding systemic regulatory roles. Future research should prioritize multi-omics integration, isoform-specific inhibitors, and longitudinal clinical validation. Interdisciplinary efforts to decode m$^6$A's multifaceted regulation may advance precision diagnostics and mechanism-based therapies, ultimately improving renal disease management.

## INTRODUCTION

Kidney diseases, encompassing a spectrum of conditions such as acute kidney injury (AKI), chronic kidney disease (CKD), diabetic kidney disease (DKD), renal cell carcinoma (RCC), and lupus nephritis (LN), represent a growing global health burden. It is anticipated that chronic kidney disease on its own affects a significant portion, more than 10%, of the world's adult population, contributing to millions of fatalities and the loss of tens of millions of years of life annually (*GBD 2021 Risk Factors Collaborators, 2024*). The impact of these diseases extends beyond individual suffering, imposing significant demands on healthcare systems worldwide. Hence, gaining a comprehensive understanding of renal disease mechanisms and formulating novel treatment approaches is of paramount importance.

The recent surge in epigenetic studies has shed considerable light on the pivotal role that post-transcriptional alterations, particularly m$^6$A methylation, play in orchestrating a myriad of biological processes (*Dominissini et al., 2012*). Emerging pan-disease analyses confirm m$^6$A's fundamental regulatory significance across non-communicable pathologies (*Khan et al., 2024c*), but its kidney-specific manifestations remain incompletely characterized-a gap this review aims to address. Emerging evidence from research in RNA epigenetics demonstrates that m$^6$A methylation serves as a key modulator of post-transcriptional regulation. This dynamic epigenetic mechanism orchestrates three fundamental biological processes: subcellular RNA transport and localization, precursor mRNA splicing precision, and translational efficiency of mRNA transcripts (*Wang et al., 2014b*). Notably, emerging evidence from experimental models and clinical observations indicates a strong association between dysregulated m$^6$A modifications and renal pathophysiology. For instance, aberrant methylation patterns may contribute to fibrosis progression and podocyte injury in CKD. This highlights the urgency to dissect its molecular mechanisms for therapeutic targeting (*He & He, 2021*).

While prior research has partially revealed the involvement of m$^6$A methylation in renal pathologies, its mechanistic complexity and translational relevance remain incompletely defined. This review systematically evaluates two critical dimensions: (1) the molecular interplay between m$^6$A dynamics (*e.g.*, writer/eraser/reader dysregulation) and kidney disease progression, particularly in tubulointerstitial fibrosis and podocyte dysfunction;

and (2) the clinical applicability of m$^6$A-related biomarkers for early detection and targeted therapy. This analysis is tailored to researchers in nephrology and epigenetics, clinicians seeking mechanistic insights into renal pathologies, and translational scientists aiming to bridge molecular discoveries to therapeutic innovation. By synthesizing current evidence from epigenomic studies and interventional models, we propose a framework bridging fundamental m$^6$A biology to therapeutic innovation, such as CRISPR-based methylation editing or small-molecule modulator development. These insights are particularly relevant for precision medicine specialists designing biomarker-driven strategies, pharmaceutical scientists targeting m$^6$A regulators, and molecular biologists exploring RNA epigenetics in renal systems.

## SURVEY METHODOLOGY

A systematic approach was employed in this review to evaluate the therapeutic relevance of m$^6$A methylation in renal pathologies. A targeted search strategy was developed using Boolean-optimized terms combining three conceptual layers: (1) m$^6$A ("m$^6$A," "N6-methyladenosine", "RNA methylation"), and (2) renal focus ("kidney," "renal," "nephropathy," "nephritis"), and (3) pathology/therapeutic context ("mechanism," "therapeutic," "biomarker"), executed across Scopus, PubMed, Web of Science, Google Scholar, bioRxiv and medRxiv. Temporal coverage spanned from January 1970 to May 2025 to capture foundational and emerging studies, yielding 195 records from Scopus, 193 from PubMed, 512 from Web of Science, 282 from Google Scholar, 86 from bioRxiv, and 20 from medRxiv.

Following database retrieval, a two-stage screening protocol was implemented: initial triage by title/abstract relevance, followed by full-text appraisal against predefined criteria: inclusion of peer-reviewed original research, reviews, and clinical trials with mechanistic or therapeutic insights; exclusion of non-peer-reviewed editorials, opinion pieces, and conference abstracts (unless subsequently published in full). Preprints from bioRxiv/medRxiv were included but flagged as "unvalidated" in synthesis to balance emerging trends with methodological rigor. Study quality was assessed using standardized tools: SYRCLE's risk of bias checklist for preclinical studies (*e.g.*, randomization, blinding) and the NIH Quality Assessment Tool for clinical cohorts (*e.g.*, confounding control, exposure measurement). Low-quality studies (*e.g.*, sample size <10, inconsistent controls) were excluded ($n = 19$). Data extraction focused on m$^6$A regulators (writers/erasers/readers), disease-specific pathways, and therapeutic validations. Critical data elements-aututhor affiliations, temporal trends, methodological approaches, and translational findings-were systematically cataloged using Zotero. To mitigate selection bias, cross-referencing between primary studies and meta-analyses was performed, supplemented by expert consultations to address knowledge gaps. Dynamic updates integrated emerging evidence, maintaining review currency without compromising methodological transparency.

## m$^6$A METHYLATION OVERVIEW

m$^6$A methylation represents a fundamental epigenetic modification mechanism that orchestrates post-transcriptional gene regulation. This reversible modification, which

predominantly occurs at adenosine residues within RNA molecules, is facilitated through the intricate process of enzymatic catalysis. These enzymes have delineated a tripartite regulatory system comprising three functional protein groups that collectively maintain $m^6A$ homeostasis. This intricate system involves methyltransferases, which are commonly referred to as "writers" due to their role in adding methyl groups to specific nucleotides in RNA molecules. Additionally, the system comprises demethylases, also known as "erasers," which serve to remove the methyl groups added by the writers, thereby counteracting their effects. Moreover, the system encompasses recognition proteins, referred to as "readers," that are capable of identifying and binding to $m^6A$-modified RNA. Through this mechanism, they exert influence over a wide array of RNA processing steps and functional aspects. Together, these three groups of proteins work harmoniously to ensure the proper balance and regulation of $m^6A$ levels within cells, which is crucial for numerous biological processes (*Zaccara, Ries & Jaffrey, 2023*). The dynamic nature of this modification enables precise control over various RNA metabolic pathways. It allows for the fine-tuning of RNA processing, stability, translation, and localization, thereby influencing numerous biological processes and functions within the cell (*Zhao et al., 2021a*). The methylation process is primarily facilitated by an evolutionarily preserved methyltransferase complex, commonly referred to as "writers." This complex is principally composed of WTAP, METTL3, and METTL14 proteins. The key catalytic components of these writers demonstrate the ability to bind specifically to target sequences and coordinate spatially, thereby enabling the methylation of adenosine residues (*Liu et al., 2023*). Erasers counteract this modification, demethylases, including FTO and AlkB homologue 5 (ALKBH5), execute oxidative removal of methyl groups, establishing a dynamic equilibrium critical for adaptive cellular responses (*Chen et al., 2021*). The recognition of $m^6A$ marks by reader proteins subsequently determines the fate of RNA through the modulation of transcript stability, nuclear export efficiency, and translational activity (*Chen, Zhang & Zhu, 2019*).

The profound impact of $m^6A$ methylation extends far beyond its pivotal functions in RNA biology. It also significantly impacts disease progression. Indeed, a wealth of research has revealed its crucial involvement in the complex regulatory mechanisms underpinning a wide array of pathological states, such as cancer, neurological disorders, and cardiovascular diseases (*Chatterjee, Shen & Majumder, 2021*). These findings suggest that $m^6A$ methylation may also have implications for kidney diseases, offering a potential framework for understanding its role in renal pathologies. In cancer, $m^6A$ methylation has been shown to modulate tumor suppressor genes and oncogenes, thereby influencing cell proliferation and differentiation. Studies carried out recently, like the ones by *Qiu et al. (2023)*, have gone on to further reveal the interplay between miRNA and $m^6A$ regulators during cancer progression. In a similar vein, about neuronal function, *Geng et al. (2023)* pointed out that the FTO gene contributes to the upregulation of the apoptosis-associated protein ataxia telangiectasia mutated (ATM), especially in dopaminergic neurons. This process involves $m^6A$-mediated regulation of ATM mRNA stability, which may be disrupted under pathological conditions. These insights from other disease contexts suggest that $m^6A$ methylation could similarly regulate key processes in kidney diseases,

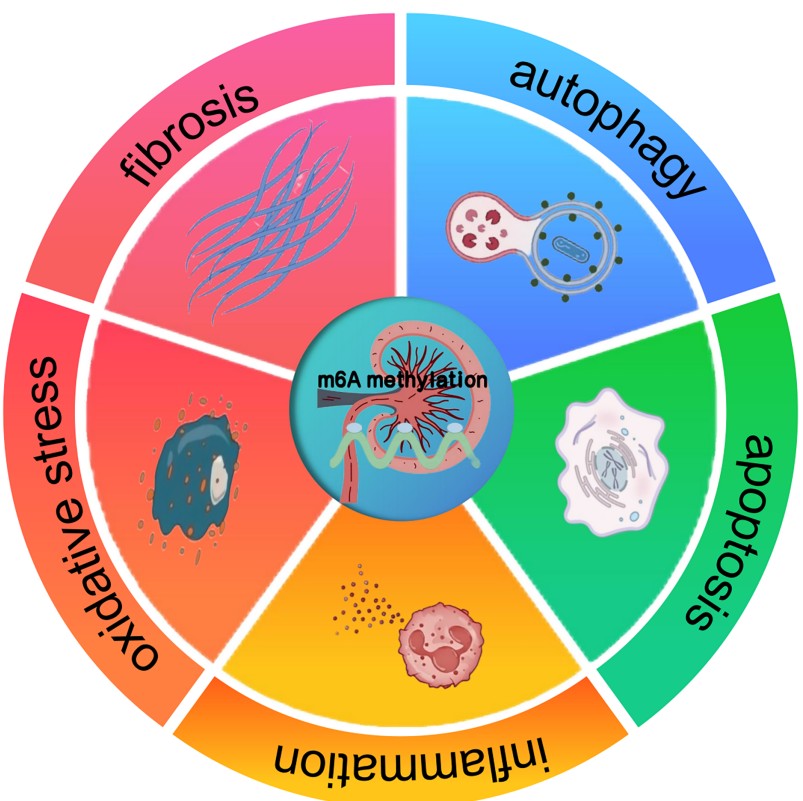

**Figure 1 The effect of m6A methylation on kidney diseases.** m⁶A methylation in kidney diseases modulates gene expression and signaling pathways, thereby influencing pathological processes such as inflammation, apoptosis, oxidative stress, autophagy, and fibrosis.

such as inflammation, apoptosis, oxidative stress, autophagy, and fibrosis (Fig. 1). The potential means by which it affects renal cell behavior and promotes the onset of diseases, including AKI, CKD, DKD, RCC, and LN, are a major focus of current research (*Guo et al., 2019*; *Xue et al., 2021*; *Qi et al., 2023*).

## m⁶A METHYLATION MODIFICATION: ENZYMES AND PROTEINS

The complex process of m⁶A methylation is orchestrated by a network of enzymes and proteins that govern its specific localization and functional roles within the transcriptome. This section explores the central components involved in m⁶A methylation, categorized as The focus is on Writers, Erasers, and Readers, and an examination of their possible contributions and the regulatory mechanisms at play in relation to kidney diseases is conducted. As depicted schematically in Fig. 2, this comprehensive regulatory network highlights the biological importance of m⁶A modification in maintaining cellular homeostasis.

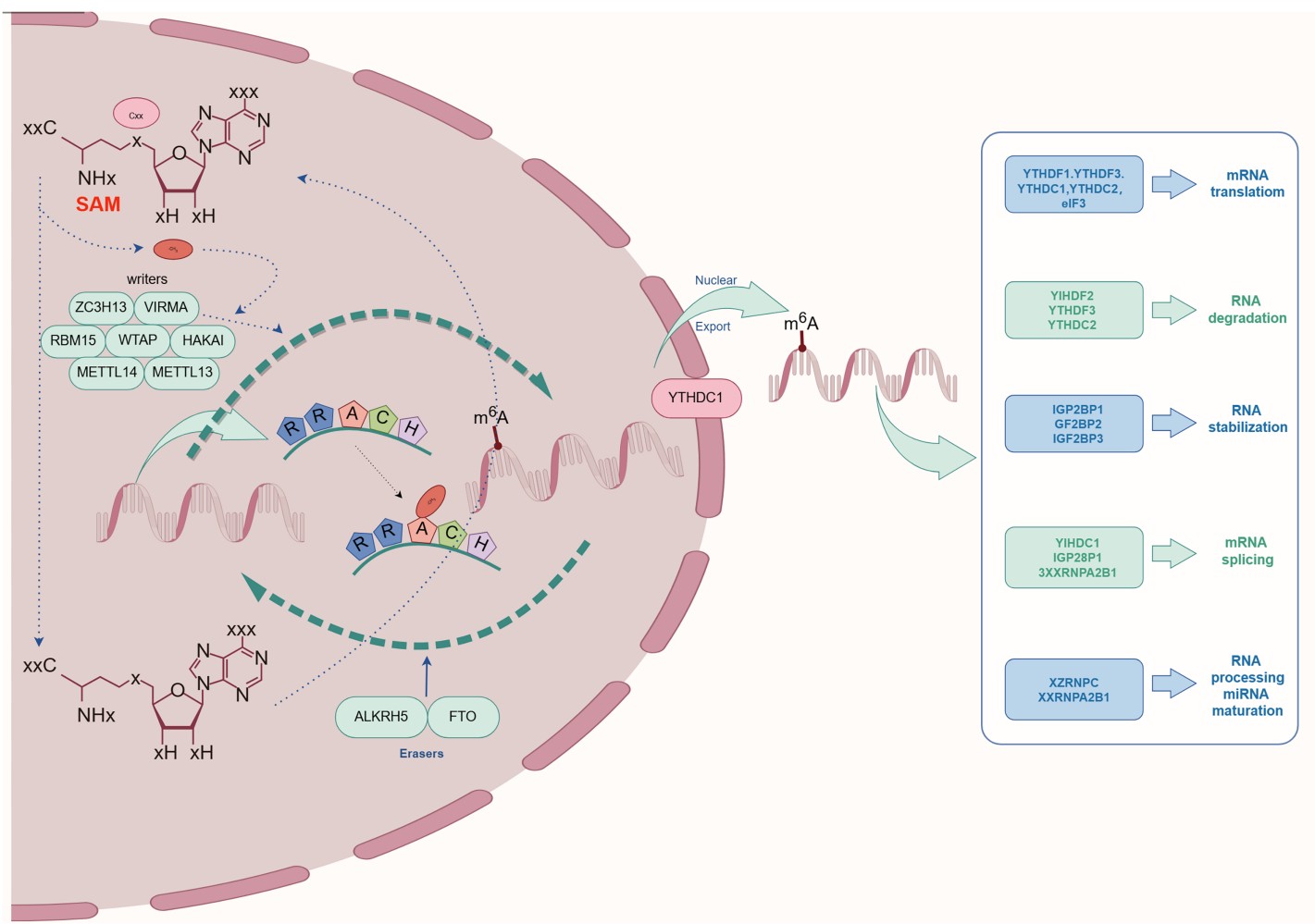

**Figure 2 The process and molecular function of m6A modification.** The m6A writers include METTL3, METTL14, METTL16, WTAP, KIAA1429, RBM15, RBM15B, ZC3H13, CAPAM, and ZCCHC4. The m⁶A erasers included FTO, ALKBH5, and ALKBH3. The m⁶A readers include the YTHDF1, YTHDF2, YTHDF3, YTHDC1, YTHDC2, HNRNPA2B1, HNRNPC, HNRNPG, IGF2BPs, including IGF2BP1/2/3), Prrc2a, HuR, and CNBP.

## Writers: the catalysts of m⁶A methylation

A particular class of methyltransferase enzymes, known as writers, is in charge of the m⁶A methylation process. These writers assemble into a sophisticated m⁶A methyltransferase complex (MTC), which functions as a pivotal hub in the intricate network of epigenetic regulation. Methyltransferase-like 3 (METTL3), methyltransferase-like 14 (METTL14) and Wilms tumor 1-associated protein (WTAP) are the key components that form the core of the MTC. This enzymatic complex governs the site-specific methylation of structurally diverse RNA substrates through its catalytic activity, establishing a crucial layer of post-transcriptional gene regulation. The regulation of this modification process not only safeguards cellular homeostasis but also ensures the fidelity of genetic information flow from transcription to protein synthesis. Disruption of these regulatory mechanisms precipitates multifaceted pathophysiological manifestations by distorting RNA-protein

interactions and compromising translational accuracy. Notably, METTL3-alternatively designated as MT-A70-acts as the catalytic subunit critical to this process, and the first-identified key writer protein (*Sorci et al., 2018*). It binds to S-adenosylmethionine (SAM) and transfers a methyl group from SAM to the sixth nitrogen atom of adenosine, thereby catalyzing m$^6$A methylation (*Bokar et al., 1997*; *Schwartz et al., 2013*; *Jiang et al., 2021*). Recent investigations have pinpointed further regulatory components situated in the MTC. These encompass viral-like m$^6$A methyltransferase proteins, RNA-binding motif proteins, zinc finger CCCH-type proteins, as well as Cbl proto-oncogene-like protein 1. Such elements exert an influence on the activity of METTL3 (*Wang et al., 2020*; *Zhu et al., 2021*; *Su et al., 2022*).

As one of the core components, METTL14 bears a strong resemblance to METTL3 in terms of its genetic makeup, yet it doesn't have the ability to catalyze reactions on its own. What it does instead is to enhance the methylation activity of METTL3, and the two join together to form a heterodimer, with METTL14 playing an irreplaceable role in the entire catalytic process (*Liu et al., 2014*; *Wang et al., 2014a*). WTAP, the third core component, lacks intrinsic methyltransferase activity but critically amplifies the catalytic efficiency of the METTL3-METTL14 heterodimer, positioning it as indispensable for methylation regulation (*Sorci et al., 2018*). At first, WTAP was found to be a splicing factor for the Wilms Tumor 1 (WT1) protein. It performs a crucial function in regulating the mammalian cell cycle and is of significant importance throughout the initial phases of embryonic development (*Shi, Wei & He, 2019*). WT1, a transcription factor specific to podocytes, is crucial for kidney growth and function (*Lee, Kim & Kim, 2014*; *Zhang, Fu & Zhou, 2019*).

Other proteins implicated in m$^6$A modification include NSUN2, a component of the SETD family, which has recently been linked to RNA methylation (*Li & Huang, 2024*). Similarly, NSUN6 (METTL16), due to its structural similarity to METTL3 and METTL14, is hypothesized to drive m$^6$A modifications on targeted RNA molecules (*Guarnacci et al., 2024*). Additionally, DNMT3A and DNMT3B, traditionally associated with DNA methylation, have shown potential RNA methylation activity, indicating a more extensive involvement in RNA modification (*Zhang et al., 2021*; *Chen et al., 2024c*). Histone methyltransferases such as H3K36, SETD6, and SETD7 are also under investigation for their potential roles in RNA methylation.

The YTH domain protein family, known as "readers" of m$^6$A modifications, influences RNA splicing, stability, and immunogenicity, contributing to immune regulation and antitumor immunity (*Ma et al., 2024a*). ZC3H13, a zinc finger protein interacting with the m$^6$A methylation complex, has an important function in m$^6$A modification (*Lin et al., 2022*). RNA-binding proteins like RBM15/RBM15B may also participate in m$^6$A modification through their interaction with the MTC (*Cao et al., 2024*; *Ma et al., 2024b*). Lastly, heterogeneous nuclear ribonucleoprotein A2B1 (HNRNPA2B1), another RNA-binding protein, is a subject of ongoing research for its potential involvement in m$^6$A modification (*Li et al., 2023b*).

## Erasers: the demethylases of m⁶A

The process of m$^6$A demethylation entails stripping away methyl groups from adenosine components within RNA molecules. This stands apart from the methylation function carried out by writer enzymes. Enzymes responsible for this function are termed erasers, with fat mass and obesity-associated protein (FTO) (*Huang et al., 2021*) and AlkB Homologue 5 (ALKBH5) (*Bokar et al., 1997*) being the most well-characterized members of this group. FTO and ALKBH5 both belong to the AlkB family, which is a group of DNA repair enzymes (*Zaccara, Ries & Jaffrey, 2019*; *Gao et al., 2024*).

FTO, widely expressed in lipid-rich tissues and the brain, contributes to regulating the alternative splicing of RUNX1T1, a key factor in lipid metabolism, and is involved in the 3′ end processing of mRNA in 293T cells (*Wang et al., 2024a*). Predominantly localized in the nucleus, FTO exhibits a unique C-terminal folding mechanism that enhances its mRNA demethylation activity, thereby having a considerable impact on gene expression (*Jia et al., 2011*). The second-identified demethylase, ALKBH5, features an A-rich motif at its N-terminus, and such a motif facilitates its presence in nuclear speckles (*Martinez De La Cruz et al., 2021*; *Qu et al., 2022*). Both FTO and ALKBH5 contain an ALKB domain in their central regions, which includes two active sites that bind Fe(II), α-ketoglutarate (α-KG), and the RNA substrate. Through its demethylation activity, ALKBH5 influences mRNA stability, splicing, and translation, as well as participates in mRNA nuclear export and processing (*Liu et al., 2023*).

## Readers: the effectors of m⁶A methylation

The fact that m$^6$A can pinpoint binding sites between RNA and methylated proteins serves as evidence for its role in modified transcripts, including mRNA (*Carroll, Narayan & Rottman, 1990*; *Song et al., 2024*). In the process of m$^6$A recognition, reader proteins specifically engage with m$^6$A-modified RNA, coordinating the recruitment of RNA-binding proteins to their designated targets and influencing the secondary structure of mRNA (*Narayan et al., 1994*; *Jiang et al., 2021*). The YTH domain protein family, which includes YTHDF1, YTHDF2, YTHDF3, YTHDC1, and YTHDC2, is made up of the three main subtypes of m$^6$A-binding proteins. It's worth noting that YTHDF1, YTHDF2, YTHDF3, and YTHDC2 are mainly found in the cytoplasm (*Xu et al., 2024*). YTHDF2 is primarily involved in regulating RNA decay, YTHDF1 and YTHDF3 govern the translational effectiveness of their target mRNAs (*Wang et al., 2015*; *Shi et al., 2017*).

YTH domain proteins are not the only ones capable of recognizing m$^6$A modifications. Additional proteins capable of binding m$^6$A, such as HNRNPA2B1, have also been identified (*Siculella et al., 2023*), heterogeneous nuclear ribonucleoprotein C, fragile X messenger ribonucleoprotein 1, and IGF2BP (*Ramesh-Kumar & Guil, 2022*), also play significant roles in this process.

Recent studies have identified Fragile X-related protein 1(FXR1) as a novel m$^6$A reader that extends the diversity of m$^6$A-mediated regulatory mechanisms. Unlike canonical readers that contain YTH or KH domains, FXR1 interacts with m$^6$A-modified RNA transcripts through an unconventional RNA-binding domain. This interaction enhances the stability and translation of target mRNAs involved in cell cycle regulation,

**Table 1 Enzymes and proteins involved in m⁶A methylation.** Summarizes the key enzymes and proteins involved in m$^6$A RNA methylation, categorized by their functional roles: methylases (*e.g.*, METTL3-METTL14 complex, WTAP), demethylases (*e.g.*, FTO, ALKBH5), and m$^6$A-binding proteins (*e.g.*, YTHDF family, IGF2BPs). Each entry includes the molecule name, its biochemical function in m$^6$A modification, and supporting references.

| Type | Molecule | Function | Reference |
|---|---|---|---|
| Methylases | Methyltransferase-like 3 (METTL3) | Using the methyl group from SAM to replace the hydrogen atom connected to the sixth nitrogen atom of adenosine | *Bokar et al. (1997)* |
| | Methyltransferase-like 14 (METTL14) | forms a complex with METTL3 to install m$^6$A on mRNA | *Sorci et al. (2018)* |
| | Methyltransferase-like 16 (METTL16) | Installs m6A onto the U6 small nuclear RNA | *Guarnacci et al. (2024)* |
| | Wilms tumor 1- associated protein (WTAP) | Regulates cellular m$^6$A levels and is part of the m6A methyltransferase complex | *Shi, Wei & He (2019)* |
| | Vir-like m$^6$A methyltransferase associated (KIAA429) | Serving as the regulatory basis for METTL3 activity | *Zhu et al. (2021)* |
| | RNA binding motif protein 15/15B (RBM15/15B) | Bind the m$^6$A methylation complex | *Cao et al. (2024)* |
| | Zinc finger CCCH domain-containing protein 13 (ZC3H13) | Anchor WTAP | *Huang et al. (2022)* |
| | Phosphorylated CTD interacting factor 1 (CAPAM) | Installs m$^6$Am at the 5′-end of most eukaryotic mRNA | *Guarnacci et al. (2024)* |
| | DNA Methyltransferase 3A/B (DNMT3A/B) | Catalytic m$^6$A modification | *Feng et al. (2024)* |
| | Histone H3 lysine 36 methyltransferase (SETD2) | SETD2 catalyses guides the deposition of m6A on nascent RNA transcripts | *Ma et al. (2024a)* |
| Demethylases | Fat mass and obesity-associated protein (FTO) | Facilitates mRNA demethylation | *Xie et al. (2019)* |
| | AlkB homologue 5 (ALKBH5) | Ensure that ALKBH5 can be localized to nuclear speckles | *Gao et al. (2024)* |
| | AlkB homolog 3 (ALKBH3) | Remove m$^6$A modification | *Gao et al. (2024)* |
| Binding proteins | YTH N6-methyladenosine RNA binding protein 1-3 (YTHDF1-3) | Translation; Phase separation; Decay | *Wang et al. (2015)*, *Shi et al. (2017)*, *Xu et al. (2024)* |
| | YTH domain containing 1 (YTHDC1) | Nuclear export; Splicing | |
| | YTH domain containing 2 (YTHDC2) | Translation | *Lv et al. (2021)*, *Chen et al. (2024d)* |
| | Insulin-like growth factor 2 mRNA binding protein 1-3(IGF2BP1-3) | Translation; Splicing; Stabilization; Decay | *Ramesh-Kumar & Guil (2022)* |
| | heterogeneous nuclear ribonucleoprotein C (HNRNPC) | Structure switching | *Liu et al. (2017)* |
| | heterogeneous nuclear ribonucleoprotein G (HNRNPG) | Splicing | *Liu et al. (2017)* |
| | heterogeneous nuclear ribonucleoprotein A2/B1 (HNRNPA2B1) | miRNA processing; Splicing | *Siculella et al. (2023)* |
| | Eukaryotic translation initiation factor 3 subunit A(eIF3) | Translation | *Gomes-Duarte et al. (2018)* |
| | cellular nucleic acid binding protein (CNBP) | Translation; stabilization | *DeAntoneo, Herbert & Balachandran (2023)* |

(Continued)

| Table 1 (continued) | | | |
| --- | --- | --- | --- |
| Type | Molecule | Function | Reference |
| | staphylococcal nuclease and tudor domain containing1 (SND1) | Stabilization | *Lan et al. (2021)* |
| | Proline rich coiled-coil 2 A (Prrc2a) | Stabilization | *Schott & Garcia-Blanco (2021)* |
| | ELAV like RNA binding protein 1 (HuR) | Stabilization | *Finan et al. (2023)* |
| | Fragile X-related protein 1 (FXR1) | Translation; stabilization | *Khan et al. (2024a, 2025)* |

inflammation, and stress responses (*Khan et al., 2024a*, *2024b*, *2025*). Importantly, FXR1 has been shown to function in both physiological contexts, such as muscle development and pathological conditions including tumorigenesis and immune regulation, suggesting its potential relevance in kidney diseases as well.

Erasers, which actively reverse m$^6$A methylation, exemplify its dynamic and reversible regulatory nature. Readers, in turn, interpret the methylation marks to regulate RNA translation and degradation, thereby fine-tuning gene expression (*Lv et al., 2021*; *Liang et al., 2022*). Dysregulation of m$^6$A readers, such as YTHDF1-3 and YTHDC1-2, can disrupt this balance, contributing to pathological conditions, including kidney diseases (*Zhang et al., 2023*).

The m$^6$A methylome and its associated proteins are dynamically regulated, underscoring their vital role in maintaining cellular homeostasis. The potential problems that can result from their improper regulation in disease states are also highlighted. The functional coordination of m$^6$A-associated enzymes and proteins is central to disease mechanisms in the kidney, modulating processes such as cellular proliferation, fibrosis, and inflammation. Dysregulation of specific Writers or Erasers, whether through overexpression or downregulation, can lead to abnormal m$^6$A methylation patterns, potentially disrupting normal renal function and exacerbating disease progression (*Li et al., 2022*). For instance, the m$^6$A reader YTHDF2 has been shown to contribute to renal fibrosis by stabilizing the mRNA of pro-fibrotic genes (*Zhao & Yang, 2024*).

Investigating the precise mechanisms by which these enzymes and proteins influence kidney pathophysiology, such as their role in modulating inflammatory responses in diabetic kidney disease (*Lan et al., 2022*), could reveal new diagnostic biomarkers and therapeutic targets. Similar observations have been previously put forward regarding cancer progression (*Qin et al., 2024*). Table 1 provides a succinct summary of the functions executed by key enzymes and proteins within the m$^6$A methylation process and their involvement in the pathogenesis of renal diseases.

## THE ROLE OF m$^6$A METHYLATION IN KIDNEY DISEASES

m$^6$A methylation, which is extensively acknowledged as a crucial mRNA post-transcriptional modification, plays a vital role in preserving physiological homeostasis and propelling the development of diseases. As summarized in Table 2, m$^6$A methylation emerges as a central regulator of renal pathophysiology across diverse disease

**Table 2 m6A methylation in kidney diseases: mechanism-clinical correlations.** This comprehensive table synthesizes m6A RNA methylation pathways across seven kidney diseases, detailing 28 molecular mechanisms (writers/erasers/readers interactions) and their 19 clinical correlations (epidemiological patterns, diagnostic biomarkers, and therapeutic implications).

| Disease | Mechanistic pathways | Clinical implications | Reference |
|---|---|---|---|
| **Acute kidney injury (AKI)** | | | |
| Drug-induced | Cisplatin<br>• Cisplatin enhances m6A methylation *via* METTL3/FTO upregulation<br>• Activates p53→Bax/Bcl-2/Caspase3 pathways | Clinical burden<br>• Responsible for nearly 25% of AKI cases<br>• Predominantly causes acute tubular necrosis | *Zhou et al. (2019)* |
| Ischemia-reperfusion | Knockout models<br>• ALKBh5 knockout improves renal function<br>• Increases Ccl28 mRNA stability*via* m6A modification | Therapeutic target<br>• 78.6% mortality in ICU patients<br>• Hydralazine prevents fibrosis in models | *Chen et al. (2023)* |
| Sepsis-associated | Ferroptosis pathways<br>• Seven key genes including Hmox1, Spp1, Socs3<br>• IGF2BP1 up-regulates MIF *via* E2F1 modification | Patient statistics<br>• Affects 35–50% of sepsis patients<br>• Mortality rate reaches 35% | *Liu et al. (2022)*, *Mao et al. (2023)* |
| **Chronic kidney disease (CKD)** | | | |
| Progression | UUO models<br>• Global m6A reduction in UUO models<br>• METTL3/14 downregulation and FTO upregulation<br>• Genistein activates ALKBH5 reducing m6A | Biomarkers<br>• Affects 10% of adults globally<br>• Peripheral blood leukocytes show reduced m6A<br>• YTHDF1↑ correlates with YAP↑ in fibrotic kidneys | *Su et al. (2022)*, *Liu et al. (2023c)* |
| **Diabetic kidney disease (DKD)** | | | |
| Podocyte injury | IGF2BP2-dependent<br>• METTL3 boosts TIMP2 m6A *via* IGF2BP2<br>• Activates Notch3/4 signaling<br>• AAV9-METTL3 silencing reduces proteinuria | Patient podocytes<br>• Impacts 50% of type 2 and 33% of type 1 diabetes<br>• METTL3↑ in DKD podocytes correlates with damage | *Jiang et al. (2022)*, *Wu et al. (2024)* |
| Tubular injury | PTEN/PI3K/Akt<br>• Regulates m6A-modified TUG1 lncRNA<br>• Activates MAPK/ERK signaling<br>• Suppresses PTEN → Activates PI3K/Akt | HRGECs models<br>• METTL14↑ in DN patients and HRGECs<br>• Overexpression amplifies ROS/TNF-α/IL-6<br>• Silencing attenuates inflammatory mediators | *Xu et al. (2021)*, *Zheng et al. (2023)* |

| Disease | Mechanistic pathways | Clinical implications | Reference |
|---|---|---|---|
| **Renal cell carcinoma (RCC)** | | | |
| METTL3 pathways | Metastasis<br>• Hypoxic conditions elevate METTL3 via HIF-1α<br>• Facilitates PLOD2 upregulation<br>• Drives tumor proliferation | TCGA/GEO<br>• Overexpression correlates with poor prognosis<br>• Distinct m6A signatures inTumor vs Normal tissue | Chen et al. (2024a) |
| **Lupus nephritis (LN)** | | | |
| Pathogenesis | Subtype-specific<br>• Two m6A modification subtypes<br>• IMP2 regulates C/EBPβ/δ stability<br>• IGF2BP2 stabilizes inflammatory transcription factors | Prognostic biomarkers<br>• Seven m6A markers correlate with renal dysfunction<br>• Targets address poor drug specificity | Slivka et al. (2019), Bechara et al. (2021) |
| **Obstructive nephropathy (ON)** | | | |
| Fibrosis | Fibrosis mechanisms<br>• METTL3-dependent induction of miR-21<br>• TGF-β elevates METTL3 levels<br>• FTO stabilizes RUNX1 mRNA viademethylation | Therapeutic potential<br>• Genistein restores ALKBH5 expression<br>• FTO silencing alleviates pathological processes | Yi et al. (2024), Jung et al. (2024), Wang et al. (2024a) |
| **Other renal pathologies** | | | |
| Autosomal dominant polycystic kidney disease (ADPKD) | Methionine/SAM<br>• METTL3 upregulation correlates with cystogenesis<br>• Elevated methionine/SAM enhance METTL3 activity<br>• Stabilizes c-Myc and Avpr2 transcripts | Murine models<br>• METTL3 deletion attenuates cyst expansion<br>• Dietary methionine restriction shows potential | Ramalingam et al. (2021) |
| Alcohol-induced nephropathy | Inflammatory pathways<br>• Suppressed FTO expression elevates m6A modification<br>• YTHDF2-mediated recognition of PPAR-α mRNA<br>• Activates NLRP3 inflammasomes and NFκBsignaling | Therapeutic targets<br>• Exacerbates inflammatory renal injury<br>• Potential for FTO/YTHDF2 targeting | Yu et al. (2021) |
| Focal segmental glomerulosclerosis (FSGS) | Glomerular dysfunction<br>• METTL14-mediated m6A modifications<br>• Suppresses Sirt1 expression<br>• Nephroprotective deacetylase downregulation | Clinical manifestations<br>• Exacerbates proteinuria and glomerulosclerosis<br>• Observed in murine models and human podocytes | Lu et al. (2021) |

contexts. This particular modification is deeply interwoven with a host of signaling pathways. These pathways engage in intricate interactions and exert mutual influences in a complex manner, thereby having an impact on a broad spectrum of biological processes and disease-associated pathways.

## Acute kidney injury

This section primarily draws on preclinical data from animal models and *in vitro* experiments to explore the role of m$^6$A modification in acute kidney injury. While these studies provide valuable mechanistic insights, further clinical validation is needed to confirm their applicability in human disease.

AKI is a medical condition characterized by a swift and significant decline in the kidneys' ability to effectively filter waste and maintain fluid balance, marking a crucial disruption in renal function, often leading to CKD and, in severe cases, end-stage renal disease (ESRD) (*Ronco, Bellomo & Kellum, 2019*). A complete understanding of the molecular mechanisms underlying AKI, particularly during its initial phases, is critical for advancing targeted therapeutic strategies. Emerging research points to the participation of m$^6$A modification in the process of acute kidney injury pathogenesis (*Mao et al., 2023*). Mechanistically, m$^6$A modification participates in AKI pathogenesis through three key etiologies: nephrotoxic drugs (*Mao et al., 2023*), sepsis (*Poston & Koyner, 2019*), and ischemia/reperfusion (I/R) injury (*Chen et al., 2023*).

Clinically, drug-induced kidney injury is responsible for nearly 25% of AKI cases, predominantly due to damage to renal tubular epithelial cells, resulting in acute tubular necrosis (*Perazella & Rosner, 2022*). Mechanistically, cisplatin a common nephrotoxic agent, accumulates in renal tubular cells and triggers multiple intracellular stress responses, such as DNA damage, oxidative stress, mitochondrial dysfunction, and endoplasmic reticulum stress (*Tang et al., 2023*). In models of cisplatin-induced AKI, an increase in the total RNA m$^6$A methylation content within kidney tissues has been detected, along with variations in the expression of key regulatory factors, such as METTL3, METTL14, WTAP, FTO, and ALKBH5 (*Li et al., 2021*). Further studies by *Zhou et al. (2019)* revealed that cisplatin enhances m$^6$A methylation by suppressing FTO while upregulating METTL3/14, leading to p53 activation and subsequent Bax/Bcl-2/Caspase3-mediated apoptosis. Therapeutically, these findings suggest that targeting METTL3 or FTO could mitigate cisplatin nephrotoxicity. A key factor leading to AKI is ischemia-reperfusion injury (IRI), with ischemic and septic acute tubular necrosis being major contributors. These conditions are associated with a high mortality rate of 78.$^6$% among hospitalized and intensive care unit patients (*Menon, Symons & Selewski, 2023*). Animal-based experimental studies have demonstrated that IRI involves a mechanistic interplay between ALKBH5 and CCL28 mRNA stability. Alkbh5 knockout mice exhibit reduced renal damage post-IRI due to increased CCL28 m$^6$A levels, which modulate the Treg/inflammatory cell axis (*Chen et al., 2023*). These findings indicate that targeting the ALKBH5/CCL28/Treg axis may offer a promising therapeutic approach for the treatment of AKI. Translational studies indicate that hydralazine, a demethylating agent, prevents IRI-related fibrosis in mice (*Tampe et al., 2017*), highlighting ALKBH5 inhibition as a

potential therapeutic strategy. Emerging research also implicates the YAP/TEAD1 pathway in protecting tubular epithelial cells from necroptosis and inflammation in cisplatin-induced AKI. In a recent study, TEAD1 overexpression preserved mitochondrial integrity and suppressed RIPK3/MLKL-mediated necroptosis in proximal tubular cells, leading to reduced inflammatory cytokine release (*Tran et al., 2025*). Although m⁶A levels were not directly measured, these findings raise the possibility that TEAD1–YAP signaling may intersect with the METTL3/METTL14 m⁶A machinery to coordinate mitochondrial and inflammatory gene expression in AKI. Sepsis-associated acute kidney injury (SA-AKI) represents a significant clinical challenge, as it is among the most prevalent and severe complications affecting patients with sepsis. Around 35% to 50% of patients with sepsis develop AKI, and the mortality rate for these patients can reach as high as 35% (*Peerapornratana et al., 2019*). m⁶A RNA methylation has been closely linked to SA-AKI pathogenesis. *In vitro* studies have identified seven key genes (*Hmox1*, *Spp1*, *Socs3*, *Mapk14*, *Lcn2*, *Cxcl1*, and *Cxcl12*) that play critical roles in SA-AKI-related signaling pathways (*Liu et al., 2022*). Notably, the mmu-miR-7212-5p-Hmox1 signaling mechanism is implicated in ferroptosis and underpins the pathophysiological characteristics of SA-AKI. In addition, it has been discovered that insulin-like growth factor 2 mRNA-binding protein 1 (IGF2BP1) can upregulate the expression of macrophage migration inhibitory factor (MIF) by enhancing E2F transcription factor 1 (E2F1) through m⁶A modification, which subsequently leads to podocyte apoptosis in SA-AKI (*Mao et al., 2023*).

The comprehensive results underscore the pivotal significance of m⁶A methylation and its associated regulatory elements, including METTL3, METTL14, FTO, and ALKBH5, in deciphering the initiation of AKI and in devising innovative therapeutic strategies. While research has demonstrated the detrimental effects of cisplatin and meclofenamic acid on kidney injury, the exact fundamental processes still need to be completely clarified. Therefore, a deeper exploration of m⁶A methylation's role in AKI, particularly in drug-induced kidney injury, is essential for pinpointing potential therapeutic targets and pushing forward treatment approaches.

## Chronic kidney disease

This section integrates preclinical research with limited clinical data to examine the role of m⁶A modification in chronic kidney disease. Although animal models and *in vitro* experiments have uncovered key molecular mechanisms, clinical evidence remains scarce.

CKD stands as a significant global health concern. It is estimated that approximately 10 percent of the adult population worldwide grapples with this condition. Moreover, the impact of CKD is far-reaching, with an annual death toll of 1.2 million individuals and a substantial burden of 28 million disability–adjusted life years (DALYs) (*GBD Chronic Kidney Disease Collaboration, 2020*). Clinically, the occurrence of CKD exhibits a strong positive correlation with advancing age (*Cockwell & Fisher, 2020*). Pathologically, CKD is characterized as a multifactorial process involving structural remodeling and progressive functional decline of the kidneys (*Kalantar-Zadeh et al., 2021*). Its progression is governed by a complex molecular network, with current research primarily focusing on key

pathological mechanisms such as immune-mediated inflammation, oxidative stress, fibrosis, apoptosis, and metabolic dysregulation.

Mechanistically, emerging epigenetic studies reveal that m$^6$A modification dynamically regulates CKD progression. Transcriptomic studies utilizing the unilateral ureteral obstruction (UUO) model have proven that m$^6$A modification is widely distributed in renal tissues and closely associated with disease progression (*Sun et al., 2023*). Of particular interest, renal tissues in the UUO model exhibit a global reduction in m$^6$A levels, accompanied by downregulation of methyltransferases METTL3/14 and upregulation of the demethylase FTO (*Liu et al., 2023c*). This m$^6$A loss is enriched in pathological pathways like epithelial-mesenchymal transition and fibrosis, as confirmed by GO/KEGG analyses (*Yue et al., 2018*). Therapeutically, interventions like genistein alleviate fibrosis by activating ALKBH5 to reduce m$^6$A levels highlighting the clinical potential of m$^6$A modulation.

Further mechanistic insights that YTHDF1 protein has been found to be significantly upregulated in multiple renal fibrosis models, including UUO, folic acid-induced, and ischemia-reperfusion injury models, with its pro-fibrotic effects potentially mediated through the regulation of the YAP signaling pathway. Clinically, YTHDF1 overexpression in human fibrotic kidneys correlates with YAP levels, while CKD patients exhibit reduced leukocyte m$^6$A and elevated FTO (*Liu et al., 2023c*). Beyond YTHDF1 and the YAP pathway, m$^6$A-dependent regulation of PTEN has been implicated in renal fibrosis. Myeloid-specific PTEN deficiency aggravates tubular inflammation and collagen deposition in UUO mice, while METTL14-mediated m$^6$A on PTEN mRNA enhances its stability and dampens PI3K/Akt-driven fibrotic signaling (*An et al., 2022*). Similar METTL14–PTEN crosstalk has been demonstrated in gastric cancer models, underscoring its translational potential in CKD (*Yao et al., 2021*). These findings suggest two therapeutic paradigms: 1. global inhibition of m$^6$A modification may confer renal protection, and 2. differential regulation of specific methyltransferases (*e.g.*, METTL14) may reverse renal dysfunction.

## Diabetic kidney disease

This section is based on preclinical studies and a growing body of clinical evidence. Research in animal models and human cohorts has shown altered m$^6$A marks in diabetic nephropathy patients, highlighting the potential of m$^6$A regulators as biomarkers.

Renal fibrosis is a common pathological manifestation in DKD patients, impacting roughly half of individuals diagnosed with type 2 diabetes, and affecting about one-third of those living with type 1 diabetes, ultimately leading to the progression of CKD. Mechanistically, total flavonoids of *Abelmoschus manihot* (TFA) alter METTL3-associated m$^6$A dynamics, activate the NLRP3 inflammasome, and modulate PTEN/PI3K/Akt signaling. In this way, TFA can significantly reduce podocyte injury and death under high glucose (HG) circumstances (*Liu et al., 2021b*). This discovery offers fresh perspectives on the intricate mechanisms that underlie podocyte injury triggered by DKD. Clinically, METTL3 overexpression in DKD patients' podocytes correlates with kidney damage, while its genetic deletion reduces inflammation/apoptosis in experimental models (*Chen et al.,*

2024b). Translational studies show that silencing the METTL3 gene *via* the AAV9 vector alleviates proteinuria and pathological tissue damage in streptozotocin (STZ)-induced diabetic mice and db/db mice (*Wu et al., 2024*), highlighting therapeutic potential. Concurrently, METTL3-driven m⁶A hypermethylation exacerbates podocyte injury through the TIMP2/Notch axis (mechanistic basis); clinically, this pathway may be pharmacologically targeted by METTL3 inhibitors currently in preclinical development (*Jiang et al., 2022*).

During the progression of DN, endoplasmic reticulum stress (ERS) is vital in causing apoptosis of renal tubular epithelial cells. It has been highlighted by recent studies that the lncRNA taurine-upregulated gene 1 (TUG1), with its initial connection to retinal development, is a participant in the pathogenesis of DKD (*Ageeli Hakami, 2024*). Mechanistic insights further identify METTL14's dual roles. On one hand, METTL14 has been shown to regulate m⁶A and modify TUG1, thereby activating the MAPK/ERK signaling pathway, which promotes the death of renal tubular epithelial cells and exacerbates ERS, accelerating the progression of DKD (*Zheng et al., 2023*). On the other hand, under high-glucose conditions, METTL14 expression and m⁶A methylation levels decrease. Additionally, METTL14-mediated PTEN inhibits the activation of the PI3K/Akt pathway, leading to the upregulation of HDAC5 in renal tubular cells under hyperglycemic conditions, followed by epithelial-to-mesenchymal transition (EMT) (*Xu et al., 2021*). Interestingly, METTL14 expression is upregulated in renal tissues of diabetic nephropathy patients and in human renal glomerular endothelial cells (HRGECs) exposed to high-glucose conditions, suggesting its critical role in exacerbating renal inflammatory responses and oxidative stress (*Fu et al., 2024*). Overexpressing METTL14 in HRGECs amplifies reactive oxygen species (ROS), TNF-α, and IL-6 production, triggering enhanced cellular apoptosis. Conversely, METTL14 silencing attenuates these inflammatory mediators and apoptotic responses (*Yang et al., 2024a*). Additionally, METTL14 influences the expression of α-klotho through m⁶A modification, deteriorating kidney function, and inflammatory responses in db/db mice. Clinically, METTL14-mediated α-klotho suppression worsens DKD, though α-klotho partially compensates, suggesting combination therapies. Under hyperglycemic conditions, overexpression of WTAP markedly amplifies NLRP3 inflammasome activation, driving the secretion of pro-inflammatory cytokines such as IL-1β and IL-18, which aggravates renal injury (*Huang et al., 2024*). The insights gleaned from the study underscore the crucial role played by m⁶A modification, orchestrated by WTAP, in cellular signaling pathways. This suggests its significant therapeutic potential in the context of DN. Notably, WTAP deficiency in pancreatic β-cells induces metabolic disturbances, as evidenced by hyperglycemia and impaired insulin secretion in β-cell-specific WTAP knockout (Wtap-betaKO) mice. Mechanistic studies reveal that WTAP loss reduces m⁶A modification levels, impairing the expression of transcription factors and genes essential for insulin production (*Li et al., 2023a*). This dual regulatory function-linking WTAP to both renal pathology and β-cell dysfunction-emphasizes its systemic impact on diabetes progression. In parallel, histone deacetylase 5 (HDAC5) has been implicated in DN pathogenesis. *Xu et al. (2021)* demonstrated that HDAC5 expression was upregulated in both diabetic and UUO models.

Furthermore, suppression of HDAC5 mitigates the progression of hyperglycemia-driven EMT within renal tubular cells. HDAC5 modulates transforming growth factor β1 (TGF-β1) activity *via* the PI3K/Akt pathway, thereby promoting renal fibrosis under high-glucose conditions. These findings position HDAC5 as a viable target for mitigating fibrotic damage in DN.

Collectively, m$^6$A regulators (METTL3, METTL14, WTAP) orchestrate DKD progression through NLRP3 activation, oxidative stress, and fibrosis. Clinical correlations in patient tissues and therapeutic efficacy in preclinical models (*e.g.*, TFA, AAV9-METTL3 silencing) validate their translational relevance. Targeting these hubs may disrupt pathogenic networks while addressing compensatory mechanisms like α-klotho restoration.

## Renal cell carcinoma

This section relies on preclinical data and clinical observations. Studies in animal models and human renal cancer tissues have identified significant alterations in m$^6$A regulators, which may serve as prognostic biomarkers.

Emerging studies demonstrate that m$^6$A RNA modification, orchestrated by diverse regulators, critically influences RCC pathogenesis by modulating oncogenic or tumor-suppressive pathways. Mechanistically, METTL3 overexpression in advanced RCC is driven by HIF-1α in hypoxia, facilitating m$^6$A-dependent PLOD2 upregulation to drive tumor proliferation and metastasis (*Chen et al., 2024a*). This positions METTL3 inhibition as a potential therapeutic strategy.

Conversely, METTL14 downregulation reduces m$^6$A modification of BPTF mRNA, enhancing its stability to promote glycolysis and metabolic reprogramming (*Zhang et al., 2021*). While METTL14 deficiency activates the ATP-P2RX$^6$-Ca2+-ERK1/2-MMP9 axis through P2RX$^6$ mRNA stabilization, accelerating invasion (*Gong et al., 2019*). Clinically, these molecular alterations correlate with poor prognosis in RCC patients, with TCGA/GEO datasets confirming distinct m$^6$A regulatory signatures between tumor and normal tissues that predict progression trajectories (*Ying et al., 2021*; *Wang et al., 2024c*).

WTAP, another methyltransferase complex component, modifies m$^6$A patterns on lncRNAs to influence RCC progression. Further mechanistic insights reveal WTAP-mediated m$^6$A enrichment on TEX41 lncRNA triggers YTHDF2-dependent degradation, altering SUZ12-associated histone methylation and HDAC1 expression (*Zhou et al., 2024*). Therapeutically, experimental manipulation using dCas13b-M3M14 fusion proteins to overexpress m$^6$A-modified MUC15 and HRG suppresses RCC metastasis (*Gan et al., 2021*), suggesting targeted epitranscriptome editing as a viable strategy. Prognostically, the miR-501-3p-CDK2 axis amplifies WTAP expression, stabilizing S1PR3 mRNA *via* IGF2BP to activate PI3K/Akt signaling, correlating with reduced survival (*Ying et al., 2021*; *He et al., 2021*). While demethylases like FTO and ALKBH5 show dysregulation, their controversial roles necessitate further validation as clinical biomarkers (*Strick et al., 2020*; *Guimarães-Teixeira et al., 2021*). Collectively, these findings position METTL3 inhibition and m$^6$A-targeted editing as promising clinical interventions against RCC progression.

## Lupus nephritis

This section incorporates clinical data from human cohorts, including elevated METTL3 expression in LN biopsy tissues and PBMCs from systemic lupus patients, supporting the relevance of $m^6A$ dysregulation in this disease.

LN, a notable complication associated with systemic lupus erythematosus (SLE), comes into being as a result of immune system dysregulation and persistent inflammatory processes. At the core of its pathogenesis lies the formation of endogenous immune complexes. Such complexes set in motion complement cascades and launch inflammatory pathways, ultimately leading to the damage of renal parenchymal cells. Emerging research emphasizes the interplay between immune cell infiltration and environmentally triggered epigenetic alterations in LN progression (*Yu et al., 2022*; *Li et al., 2024*). Mechanistically, $m^6A$ RNA methylation heterogeneity defines two subtypes (1/2) with distinct expression patterns of regulators including METTL3, WTAP, YTHDC2, YTHDF1, FMR1, and FTO (*Zhao et al., 2021b*). These subtypes exhibit divergent immune microenvironment profiles and pathway activation. Clinically, seven $m^6A$-related markers correlate strongly with renal dysfunction, offering prognostic value and guiding subtype-specific immunotherapy.

Further mechanistic insights reveal the role of $m^6A$ readers in cytokine signaling modulation. For instance, IMP2, an $m^6A$-binding protein, governs autoimmune inflammation by post-transcriptionally regulating C/EBPβ and C/EBPδ stability in response to IL-17 and TNFα (*Bechara et al., 2021*). Therapeutically, this regulatory axis influences cytokine-driven renal injury, positioning IMP2 as a promising intervention target for LN. Similarly, targeting the IGF2BP2/$m^6A$-epigenetic axis has been shown to mitigate autoimmune antibody-induced glomerular damage by stabilizing transcription factors critical for inflammatory signaling, and addresses clinical challenges like poor drug specificity and limited efficacy of conventional treatments (*Slivka et al., 2019*). Collectively, these advances position $m^6A$ regulators as precision therapeutic targets against LN.

## Obstructive nephropathy

This section is based on preclinical studies and clinical observations. Animal models and human tissue samples have shown that $m^6A$ regulatory factors drive fibrotic progression in obstructive nephropathy.

Ureteral obstruction disrupts renal physiology through hemodynamic disturbances, impaired glomerular filtration, and metabolic dysregulation, culminating in structural pathologies such as renal fibrosis. The UUO model serves as a standard experimental tool for investigating tubulointerstitial fibrosis, enabling multi-level exploration of fibrotic mechanisms spanning molecular, genomic, and cellular domains (*Nørregaard et al., 2023*). Mechanistically, $m^6A$ regulatory factors critically drive fibrotic progression in obstructive nephropathy (ON) (*Liu et al., 2021a*; *Jung et al., 2024*). The up-regulation and activation of METTL3 and other pivotal methyltransferases are critical for regulating the expression of fibrosis-related genes and propelling the progression of fibrotic pathogenesis. Fibrosis-associated miRNAs, including miR-145 and miR-21, exhibit dynamic interactions with $m^6A$ modifications to regulate key signaling pathways. Notably, METTL3-dependent induction of miR-21 activates the miR-199a-3p/Par4 axis, amplifying fibrotic cascades (*Yi*

*et al., 2024*). TGF-β stimulation of HK-2 cells elevates METTL3 levels and subsequent m$^6$A modifications, upregulating C/EBP transcription factors. METTL3 inhibition attenuates C/EBPβ and C/EBPδ expression, confirming m$^6$A-dependent regulation of these transcriptional regulators. Functional studies demonstrate that C/EBPβ and C/EBPδ mediate TGF-β-driven EMT and fibrotic remodeling (*Jung et al., 2024*). Clinically, these molecular pathways present intervention opportunities, as evidenced by ALKBH5's role in UUO models, where its reduced expression correlates with elevated m$^6$A levels and exacerbated fibrosis. Therapeutically, genistein administration restores ALKBH5 expression, mitigating m$^6$A hypermethylation and fibrosis (*Ning et al., 2020*), paralleling outcomes from genetic ALKBH5 manipulation studies.

Further mechanistic insights reveal that FTO promotes fibrogenesis by modulating pro-fibrotic gene methylation, with FTO silencing shown to alleviate UUO- and TGF-β1-induced inflammation, apoptosis, and impaired autophagy (*Zhang et al., 2022*; *Yang et al., 2023*). At the molecular level, transcriptomic profiling identifies RUNX1 as a key downstream effector, where FTO stabilizes RUNX1 mRNA *via* demethylation to activate the PI3K/AKT pathway and propagate fibrotic signaling (*Wang et al., 2024a*) Clinically, this FTO-RUNX1 axis offers a promising therapeutic target, as pharmacological FTO inhibition reverses pathological RUNX1 upregulation in experimental models. Collectively, these findings position METTL3, ALKBH5 and FTO as mechanistically validated targets with translational potential for mitigating obstruction-induced renal fibrosis (*Wang et al., 2024a*).

## Other kidney diseases

This section includes preclinical data and clinical observations. Studies in animal models and human cohorts have identified specific m$^6$A regulators associated with various renal pathologies, such as alcohol-induced nephropathy and autosomal dominant polycystic kidney disease.

Emerging evidence highlights the regulatory significance of m$^6$A methylation across diverse renal pathologies through distinct mechanistic pathways with clear clinical implications. In alcohol-induced nephropathy, suppressed FTO expression elevates m$^6$A modification of PPAR-α mRNA *via* YTHDF2-mediated recognition, activating NLRP3 inflammasomes and NFκB signaling to exacerbate inflammatory renal injury (*Yu et al., 2021*). Clinically, this pathway identifies FTO/YTHDF2 as potential therapeutic targets for alcohol-related kidney damage. Parallel investigations in autosomal dominant polycystic kidney disease (ADPKD) reveal METTL3 upregulation correlating with cystogenesis, where mechanistically elevated methionine and SAM levels enhance METTL3 activity, promoting m$^6$A-dependent stabilization of c-Myc and Avpr2 transcripts to facilitate cyst growth through c-Myc-driven proliferation and cAMP pathway activation (*Ramalingam et al., 2021*). Therapeutically, transgenic METTL3 deletion attenuates cyst expansion in murine models, while dietary methionine restriction demonstrates clinical potential by disrupting these metabolic-epigenetic interactions. In focal segmental glomerulosclerosis (FSGS), mechanistically dysregulated m$^6$A dynamics contribute to glomerular dysfunction through METTL14-mediated m$^6$A modifications that suppress Sirt1 expression-a

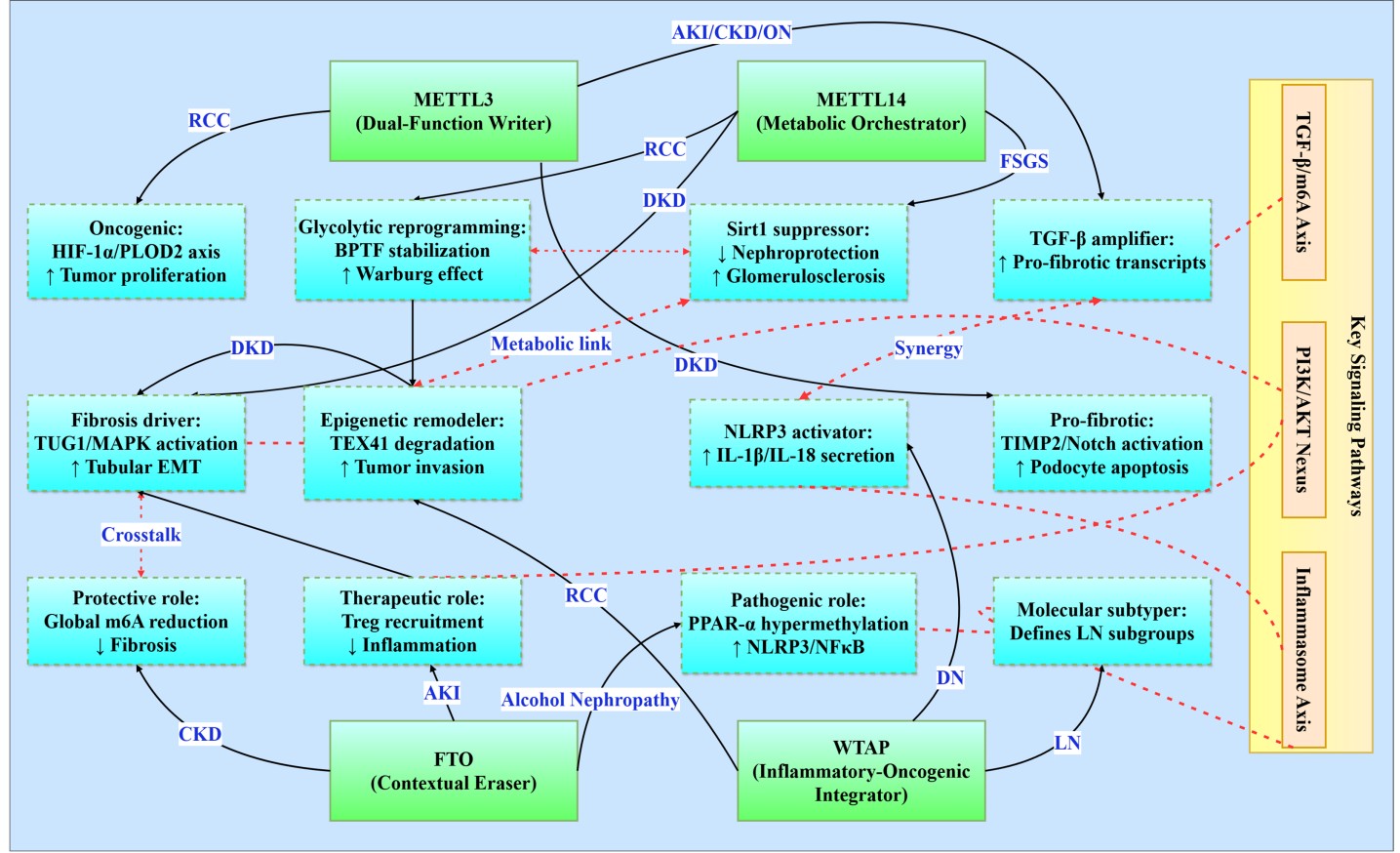

**Figure 3 Functional networks of m6A regulators in renal pathologies.** Illustrates the functional networks of m⁶A regulators in renal pathology. It features core regulators (METTL3, METTL14, FTO, and WTAP) and their respective networks involved in various renal conditions (*e.g.*, RCC, DKD, CKD). Additionally, it highlights interactions between key signaling pathways (TGF-β/m⁶A Axis, PI3K/AKT Nexus, Inflammasome Axis) and the functional networks. Solid line: direct regulation, dashed line: path interaction.

nephroprotective deacetylase-in both murine models and human podocytes (*Lu et al., 2021*). Clinically, this METTL14/Sirt1 axis exacerbates proteinuria and glomerulosclerosis, positioning METTL14 modulation as a therapeutic strategy.

Collectively, these disease-specific regulatory networks highlight distinct mechanistic pathways: alcohol-associated injury involves FTO/YTHDF2-mediated inflammatory activation, ADPKD centers on METTL3-driven metabolic-epigenetic crosstalk, and FSGS links to METTL14-dependent Sirt1 downregulation. Such insights advance renal disease etiology understanding while clinically validating m⁶A modifiers as therapeutic targets across nephropathies.

## Integrated signaling networks across renal pathologies

Epitranscriptomic regulation by m⁶A methylation interfaces with conserved signaling pathways that transcend traditional disease classifications (Fig. 3). This integrative analysis reveals four principal regulatory networks central to renal pathophysiology (Table 3), with implications for targeted therapeutic interventions.

**Table 3 Cross-disease signaling pathways in renal pathologies.** Delineates conserved signaling pathways modulated by m⁶A methylation across renal pathologies, highlighting three critical dimensions: molecular mechanisms, disease-specific manifestations, and therapeutic targeting strategies. Four principal networks emerge as central regulators.

| Pathway | Disease-specific manifestations | Therapeutic targeting |
|---|---|---|
| TGF-β/m⁶A Axis Fibrosis regulation | AKI: TGF-β↑ → METTL3↑ → Fibrosis | TGF-β inhibitors (*e.g.*, pirfenidone) |
| | CKD: METTL3 → TGF-β persistence | |
| | DKD: METTL3-TGF-β → Podocyte injury | |
| | ON: TGF-β → METTL3 → EMT | |
| PI3K/AKT Nexus Cell survival/proliferation | DKD: HDAC5 activation → Fibrosis | PI3K inhibitors (*e.g.*, LY294002) |
| | ON: FTO-RUNX1 → Fibrosis | |
| | RCC: WTAP-S1PR3 → Proliferation | |
| NLRP3 inflammasome inflammatory activation | DKD: WTAP → NLRP3 → IL-1β/IL-18 | NLRP3 inhibitors (*e.g.*, MCC950) |
| | Alcohol Nephropathy: FTO☒ → PPAR-α m⁶A↑ | |
| | SA-AKI: IGF2BP1-MIF inflammation | |
| Metabolic reprogramming energy metabolism shift | ADPKD: METTL3-c-Myc → Glycolysis | Metabolic modulators (*e.g.*, 2-DG) |
| | DKD: METTL14-TUG1 → MAPK signaling | |
| | RCC: METTL14-BPTF → Glycolysis | |

*Core signaling pathways*

The TGF-β/m⁶A axis establishes a bidirectional relationship where TGF-β signaling induces METTL3 expression in acute kidney injury (enhancing pro-fibrotic transcript modification), perpetuates CKD progression through Smad3 phosphorylation, exacerbates diabetic podocyte injury *via* TIMP2/Notch activation, and drives obstructive nephropathy EMT through C/EBPβ-mediated mechanisms (*Wang et al., 2022*; *Jung et al., 2024*; *Yang et al., 2024c*). Concurrently, the PI3K/AKT nexus integrates metabolic-proliferative cascades across pathologies: HDAC5 activates PI3K/AKT for fibrotic transformation in diabetic kidney disease, FTO stabilizes RUNX1 mRNA to activate PI3K/AKT-mediated fibrosis in obstructive nephropathy, and WTAP-mediated S1PR3 stabilization stimulates tumor proliferation in renal cell carcinoma (*Wang et al., 2024a*). In parallel, the YAP/TEAD1-m⁶A axis emerges as a conserved regulator in AKI by modulating necroptosis and inflammatory gene expression *via* m⁶A-mediated transcript stabilization (TEAD1 overexpression reduces cell death and cytokine release) (*Tran et al., 2025*).

The inflammasome activation axis centers on NLRP3 as a convergent inflammatory mechanism, with WTAP-dependent amplification in diabetic nephropathy promoting IL-1β/IL-18 secretion, FTO suppression inducing PPAR-α hypermethylation in alcohol-induced nephropathy, and IGF2BP1-MIF priming in sepsis-associated AKI (*Yu et al., 2021*; *Lan et al., 2022*; *Mao et al., 2023*). Moreover, METTL14-mediated m⁶A modification of PTEN mRNA integrates into the PI3K/Akt network to suppress fibrotic and inflammatory responses, as evidenced by worsened renal inflammation in PTEN knockout models and restored PTEN stability upon METTL14 activity (*An et al., 2022*). Complementing these, metabolic-epigenetic circuitry underlies disease-specific adaptations: METTL3 stabilizes c-Myc to enhance glycolytic flux in polycystic kidney disease, METTL14 regulates TUG1 lncRNA to activate MAPK signaling in diabetic

tubular injury, and METTL14 deficiency promotes Warburg-like remodeling *via* BPTF stabilization in renal carcinoma (*Ramalingam et al., 2021*; *Zhang et al., 2021*; *Zheng et al., 2023*).

### m⁶A regulator functional networks

m⁶A regulators exhibit context-dependent functionality within conserved frameworks. METTL3 serves as a dual-function mediator-pro-oncogenic in RCC *via* HIF-1α-PLOD2 while pro-fibrotic in diabetic nephropathy through TIMP2/Notch-while consistently amplifying TGF-β signaling across AKI/CKD/obstructive nephropathy (*Liu et al., 2020*; *Wang et al., 2022*; *Jiang et al., 2022*; *Chen et al., 2024a*). METTL14 orchestrates fibrosis through divergent mechanisms: MAPK/ERK activation *via* TUG1 regulation in diabetic kidney disease, Sirt1 suppression in focal segmental glomerulosclerosis, and BPTF-driven glycolytic reprogramming in RCC (*Zhang et al., 2021*; *Lu et al., 2021*; *Zheng et al., 2023*). FTO demonstrates paradoxical roles as a contextual protector, offering CKD protection through global m⁶A reduction yet exacerbating alcohol-induced nephropathy *via* PPAR-α hypermethylation and mediating therapeutic Treg recruitment in AKI (*Wang et al., 2020*; *Yu et al., 2021*; *Yang et al., 2024b*). WTAP integrates inflammatory-oncogenic processes, driving NLRP3 activation in diabetic nephropathy, enabling epigenetic remodeling through TEX41 degradation in RCC, and defining molecular subtypes in lupus nephritis (*Zhao et al., 2021b*; *Lan et al., 2022*; *Zhou et al., 2024*).

### Therapeutic integration

These networks position m⁶A machinery as master regulators amenable to three strategic interventions: (1) Pathway-targeted inhibition (*e.g.*, TGF-β/PI3K antagonists), (2) Context-specific regulator modulation (METTL3 inhibition in RCC *vs* activation in AKI recovery), and (3) Combined epitranscriptomic-metabolic targeting (dual METTL3/glycolytic enzyme inhibition in cystic diseases). Such approaches promise cross-disease efficacy with minimized off-target effects, advancing precision nephrology frontiers.

## Clinical evidence and translational insights

Translational efforts are increasingly incorporating human data on m⁶A regulators in kidney disease, like in LN, METTL3 expression was found to be markedly elevated. One study reported significantly higher METTL3 and ALKBH5 levels in PBMCs from systemic lupus patients, and robust METTL3 immunostaining in LN biopsy tissues, notably in infiltrating immune cells and tubules (*Liu et al., 2024*). Similarly, transcriptome analyses of human kidney biopsy cohorts have revealed dysregulated m⁶A machinery. In IgA nephropathy, for instance, nine m⁶A regulators (readers/erasers such as YTHDF2 and IGF2BP3) were significantly downregulated in patient glomeruli compared to controls (*Wang et al., 2024b*). In DKD, emerging evidence shows altered m⁶A marks in patients: diabetic nephropathy patients exhibit reduced global m⁶A levels in urine and blood, alongside decreased FTO expression (*Li & Mu, 2024*). Notably, a logistic regression model combining glomerular YTHDC1, METTL3 and ALKBH5 expression distinguished early *vs* advanced DKD, and retained predictive accuracy when applied to patients' PBMC samples

(2025). These clinical findings support the relevance of m$^6$A dysregulation in human kidney disease and hint at biomarker potential.

Beyond glomerulonephritis, m$^6$A regulators have been linked to CKD and renal cancers. In clear-cell renal cell carcinoma (ccRCC), multiple m$^6$A enzymes- including the writer METTL3 and demethylase ALKBH-are significantly overexpressed in tumor tissue *vs* normal kidney (*Chen et al., 2020*). In the same study, a two-gene signature (METTL3 and METTL14) was constructed by LASSO regression to stratify ccRCC patient survival, achieving good prognostic performance (*Chen et al., 2020*). Other CKD forms also show altered m$^6$A profiles: for instance, patient samples from ADPKD demonstrate upregulation of METTL3 in cystic kidneys. Collectively, these patient-based studies illustrate that many m$^6$A alterations observed in model systems likewise occur in human disease, bolstering their translational significance.

To date, however, interventional m$^6$A-targeted therapies in nephrology remain unrealized. A recent review emphasizes that "currently there is no drug related to RNA methylation modification that is on clinical trial for kidney diseases" (*Luan, Kopp & Zhou, 2022*). No ClinicalTrials.gov entries specifically address m$^6$A modulation in renal patients (as of mid-2025). This gap underscores the nascent stage of clinical translation in this field: mechanistic insights abound, but rigorous human validation and trials are just beginning.

To bridge mechanism and medicine, multi-center prospective studies are needed to evaluate m$^6$A regulators as biomarkers and targets. Cohorts of CKD patients (*e.g.*, diabetic nephropathy, glomerulonephritis, polycystic kidney disease) should be followed longitudinally with systematic collection of biospecimens (blood, urine, biopsy). Quantitative assays for m$^6$A levels and regulator expression (*e.g.*, LC-MS for m$^6$A/A ratio, RT-qPCR or ELISA for METTL3/FTO/ALKBH5) should be standardized and correlated with clinical outcomes (eGFR decline, proteinuria, histology). Multi-omics integration (combining m$^6$A methylome, transcriptome, proteome) in patient samples could uncover disease-driving networks and refine therapeutic targets. On the interventional front, preclinical testing of m$^6$A modulators in kidney disease models is warranted. Ultimately, early-phase trials- for instance, repurposing epigenetic drugs or novel RNA methylation inhibitors in selected patient subsets- could establish proof-of-concept. Taken together, these approaches would help translate m$^6$A biology into clinically actionable strategies in nephrology.

Patient-based analyses have begun to tie m$^6$A to renal outcomes. For example, *Zhao et al. (2021b)* identified an LN-specific m$^6$A signature (METTL3, WTAP, YTHDC2, YTHDF1, FMR1) that cleanly separated lupus nephritis biopsies from healthy kidney tissue. *Chen et al. (2020)* showed METTL3 and ALKBH5 upregulation in ccRCC tumors and derived a METTL3/METTL14 prognostic score. In DKD, *Shi et al. (2017)* found urinary m$^6$A levels fall with disease progression, and constructed a glomerular/PBMC model including METTL3 and ALKBH5 to diagnose early DKD (*Li, Xu & Li, 2025*). Such clinical reports, though still limited, illustrate the bridge between m$^6$A mechanistic work and human disease.

## TREATMENT STRATEGY

The detection of abnormal m$^6$A modification levels and dysfunctional m$^6$A-associated regulatory pathways in patients holds crucial clinical value for the diagnosis and therapeutic management of kidney diseases. Current research prioritizes identifying dysregulated m$^6$A-modifying enzymes in renal biopsies or peripheral blood mononuclear cells (PBMCs), as these alterations may represent promising diagnostic, monitoring, and prognostic biomarkers for kidney pathologies (*Li, Xu & Li, 2025*). In principle, one could imagine using m$^6$A "signatures" to stratify patients for specific treatments (analogous to oncology settings where m$^6$A patterns predict immunotherapy response), although such applications remain hypothetical in nephrology. On the therapeutic front, preclinical models demonstrate that modulating m$^6$A enzymes can affect kidney pathology. Inhibiting the m$^6$A "eraser" ALKBH5 with the small molecule IOX1 (a broad-spectrum 2-OG oxygenase inhibitor) was shown to suppress ALKBH5 activity and prevent acute kidney injury in animals (*Fang et al., 2025*). Conversely, the natural flavonoid genistein upregulates ALKBH5 in a rodent fibrosis model, lowering overall m$^6$A levels and attenuating renal fibrosis (*Seo et al., 2019*). These results illustrate that targeting m$^6$A erasers can modulate disease processes *in vivo*. Other approaches include using m$^6$A profiles to guide therapy (for instance, tailoring immunosuppressive or targeted agents based on m$^6$A-dependent pathways), though such strategies have not yet been validated clinically.

The integration of artificial intelligence (AI) technology has provided a powerful impetus for the discovery of m$^6$A modification-related drugs. In parallel, artificial intelligence and structure-based methods are emerging to aid m$^6$A-targeted drug discovery. For example, high-accuracy protein modeling (*e.g.* AlphaFold) and virtual screening have been applied to m$^6$A enzymes. A recent study used *in silico* docking with the solved ALKBH5 structure to identify a novel selective inhibitor (ALK-04) (*Fang et al., 2025*). Likewise, structure-based screening of compound libraries has yielded several m$^6$A demethylase inhibitors. *Peng et al. (2019)* screened FDA-approved drugs against the FTO (another m$^6$A demethylase) structure and identified entacapone as a competitive FTO inhibitor (*You et al., 2022*). These computational pipelines (combining predicted or experimental structures with docking) can prioritize candidate modulators of m$^6$A "writers" or "erasers" for testing in renal models. In principle, AI tools could also help predict off-target effects or optimize pharmacokinetic properties of these molecules, integrating with medicinal chemistry to accelerate development.

### Limitations of current strategies

Despite promise, significant challenges remain. Many reported inhibitors are not highly specific for a single m$^6$A enzyme. For instance, IOX1 inhibits many 2-OG dioxygenases (not just ALKBH5), raising off-target risk (*Fang et al., 2025*). Furthermore, compounds identified by virtual screens often lack selectivity or have unknown *in vivo* behavior. Natural products like genistein can have broad biological effects beyond m$^6$A modulation, and their pharmacokinetic profiles in patients are poorly characterized (*Ning et al., 2020*). Moreover, virtually all data so far come from cell lines or animal models; human validation is lacking. m$^6$A-based biomarkers and therapeutic hypotheses must be tested in large, well-controlled

patient cohorts. The idea of using m⁶A profiles to guide, say, immunotherapy or targeted drug choice in kidney disease is attractive but remains speculative without clinical evidence. In short, issues of target specificity, off-target toxicity, pharmacodynamics, and population variability have not been resolved (*Tang et al., 2024*).

**Future directions (toward clinical translation)**

To bridge these gaps, research should pursue several avenues. First, the development of highly selective m⁶A modulators is crucial. Advances in AI-driven design can help: for example, leveraging predicted structures of METTL3, FTO, ALKBH5, *etc.*, to design molecules with optimized specificity and drug-like properties. High-throughput screening (virtual and biochemical) should continue to identify and refine lead compounds, with follow-up *in vivo* testing of ADME/toxicity. Second, the clinical relevance of m⁶A markers must be validated. Large-scale studies should assess whether m⁶A enzyme levels or global m⁶A patterns in blood or biopsy correlate with disease stage, progression, or response to therapy. Third, combination strategies should be explored: it may be that partial modulation of m⁶A synergizes with existing treatments (*e.g.* RAAS inhibitors, immunosuppressants, SGLT2 inhibitors) to improve outcomes. Finally, continued integration of AI tools—such as predictive models of m⁶A target networks and patient stratification algorithms—will be important. By systematically addressing specificity, safety, and efficacy, these steps can move m⁶A-targeted approaches from bench toward bedside in renal disease.

## CONCLUSIONS

As a pivotal RNA modification, m⁶A methylation serves as a key regulatory mechanism in cellular processes. Emerging evidence underscores its clinical relevance in kidney pathophysiology, driven by advancements in detection methodologies. Aberrant expression of m⁶A-associated regulators across renal disorders highlights their therapeutic potential, offering novel avenues for targeted interventions.

Distinct disease-specific mechanisms characterize m⁶A's involvement in renal pathologies. In RCC, m⁶A patterns correlate with tumor aggressiveness and metastatic potential, positioning associated regulators as prognostic biomarkers (*Zhou et al., 2024*). AKI studies reveal WTAP, METTL3, and FTO as mediators of oxidative stress and inflammation through gene expression modulation, suggesting druggable targets (*Wang et al., 2022*; *Lan et al., 2022*; *Chen et al., 2023*). CKD progression demonstrates FTO's association with patient mortality and chronic nephritis pathogenesis (*Wang et al., 2024a*), while DKD involves METTL3/METTL14-mediated gene networks and FTO polymorphisms influencing diabetic nephropathy susceptibility (*Shukla et al., 2024*; *Li & Mu, 2024*). Renal fibrosis studies further implicate m⁶A-mediated TGF-β modulation as a critical pathway, with genetic interventions showing antifibrotic potential (*So, Yap & Chan, 2021*; *Long et al., 2024*).

Despite progress, mechanistic understanding remains incomplete. Current limitations stem from methodological heterogeneity in sample cohorts, detection platforms, and experimental designs. Most investigations focus narrowly on transcriptional regulation,

**Table 4 Abbreviations.** Compiles the abbreviations used in the article.

| Abbreviations | |
|---|---|
| m6A | N6-methyladenosine |
| AKI | Acute kidney injury |
| CKD | Chronic kidney disease |
| DKD | Diabetic kidney disease |
| RCC | Renal cell carcinoma |
| LN | Lupus nephritis |
| DN | Diabetic nephropathy |
| GFR | Glomerular filtration rate |
| ATM | Ataxia telangiectasia mutated |
| MTC | Methyltransferase complex |
| SAM | S-adenosylmethionine |
| WT1 | Wilms Tumor 1 |
| HNRNPA2B1 | heterogeneous nuclear ribonucleoprotein A2B1 |
| FTO | Fat mass and obesity-associated protein |
| ALKBH5 | AlkB homologue 5 |
| $\alpha$-KG | $\alpha$-ketoglutarate |
| FXR1 | Fragile X-related protein 1 |
| ESRD | End-stage renal disease |
| I/R | ischemia/reperfusion |
| IRI | Ischemia-reperfusion injury |
| SA-AKI | Sepsis-associated acute kidney injury |
| IGF2BP1 | insulin-like growth factor 2 mRNA-binding protein 1 |
| MIF | migration inhibitory factor |
| E2F1 | E2F transcription factor 1 |
| DALYs | disability-adjusted life years |
| UUO | Unilateral ureteral obstruction |
| TFA | Total flavonoids of *Abelmoschus manihot* |
| HG | High glucose |
| STZ | streptozotocin |
| ERS | Endoplasmic reticulum stress |
| TUG1 | Taurine upregulated gene 1 |
| EMT | Epithelial-to-mesenchymal transition |
| HRGECs | Human renal glomerular endothelial cells |
| ROS | reactive oxygen species |
| HDAC5 | Histone deacetylase 5 |
| EMT | epithelial-to-mesenchymal transition |
| TGF-$\beta$1 | Transforming growth factor $\beta$1 |
| SLE | Systemic lupus erythematosus |
| ON | Obstructive nephropathy |
| ADPKD | autosomal dominant polycystic kidney disease |
| FSGS | focal segmental glomerulosclerosis |
| PBMCs | peripheral blood mononuclear cells |
| AI | artificial intelligence |

neglecting m⁶A's systemic roles in cellular signaling and physiological homeostasis. Future directions should prioritize:

1. Elucidating pathway-specific m⁶A mechanisms through advanced multi-omics integration
2. Conducting longitudinal clinical studies to establish causal relationships
3. Developing isoform-selective inhibitors of m⁶A-modifying enzymes

The multifaceted nature of m⁶A regulation presents substantial opportunities for diagnostic innovation and therapeutic discovery. Interdisciplinary collaboration combining cutting-edge molecular techniques with clinical insights will be crucial to translate these findings into effective renal therapies. As research progresses, m⁶A-focused strategies may redefine precision medicine approaches in nephrology, ultimately improving patient outcomes through earlier detection and mechanism-driven interventions.

In this article, all the abbreviations can be found in Table 4.

## ACKNOWLEDGEMENTS

I would like to convey my heartfelt thanks to a number of people who have offered precious support and guidance during the completion of this review. Their professional guidance, patience, and encouragement have been crucial in shaping this review. Their firm belief in my capabilities and constant support have spurred me on to pursue excellence. Moreover, I would like to acknowledge the help from DeepSeek-R1, an AI that has greatly facilitated the translation and refinement of this article. Its sophisticated functions have ensured the precision and clarity of the content.

I appreciate all of your important contributions to this work.

### Funding

This work was supported by the Natural Science Foundation of Nanjing University of Chinese Medicine (No. XZR2023018). The funders had no role in study design, data collection and analysis, decision to publish, or preparation of the manuscript.

### Grant Disclosures

The following grant information was disclosed by the authors:
Natural Science Foundation of Nanjing University of Chinese Medicine: XZR2023018.

### Competing Interests

The authors declare that they have no competing interests.

### Author Contributions

- Shaowen Guo conceived and designed the experiments, performed the experiments, analyzed the data, prepared figures and/or tables, authored or reviewed drafts of the article, and approved the final draft.

- Wenjun Wang analyzed the data, prepared figures and/or tables, and approved the final draft.
- Gaopan Lv analyzed the data, prepared figures and/or tables, and approved the final draft.
- Yun Ling conceived and designed the experiments, authored or reviewed drafts of the article, and approved the final draft.
- Meifeng Zhu conceived and designed the experiments, authored or reviewed drafts of the article, and approved the final draft.

## Data Availability

This is a review article, and as such, there is no original data.

## Supplemental Information

Supplemental information for this article can be found online at http://dx.doi.org/10.7717/peerj.19940#supplemental-information.

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
