# Peer review of "The role of N6-methyladenosine (m6A) RNA methylation modification in kidney diseases: from mechanism to therapeutic potential"

_PeerJ, doi:10.7717/peerj.19940_

## Round 0.1 · original submission · Major Revisions

Reviewer 1 ·

Basic reporting

A clearer demarcation between the discussion of mechanistic pathways (e.g., writer/eraser/reader roles) and their clinical implications would improve readability and help the reader follow the narrative.
 The methods section describing the literature search and inclusion/exclusion criteria could be expanded. Clarifying the search strategy (databases used, specific keywords, time frames, and any quality assessment of studies) would enhance reproducibility and transparency.
 The exclusion criteria (e.g., "non-peer-reviewed works") are mentioned, but there is no discussion of how preprint articles or conference abstracts were handled. Given the rapid evolution of m6A research, excluding preprints might omit emerging insights.
 The discussion of signaling pathways such as the interplay between TGF-β signaling and m6A modification is robust. However, additional details on how these pathways interconnect across different renal pathologies (AKI, CKD, DN, RCC, LN) could strengthen the argument for m6A as a unifying mechanism in kidney disease.
 There appears to be an inconsistency in how different m6A regulators (for example, METTL3 versus METTL14) are described with respect to their roles in disease. Providing a concise schematic expression patterns and functional consequences across various studies would greatly aid clarity.
 It would be helpful to include a critical evaluation of the current limitations of these strategies (e.g., issues with specificity or off-target effects) and suggest potential solutions or areas for future research.
 Some sections (e.g., the parts discussing the application of AI in drug discovery) seem to introduce concepts that are not as tightly integrated into the main theme of m6A in kidney disease. It may be beneficial to better link these topics to the core discussion.
 The mechanistic role of m6A regulators (e.g., METTL3, FTO) in kidney diseases is well-summarized, the therapeutic potential section leans heavily on animal models (e.g., Alkbh5 knockout mice, STZ-induced diabetic mice) and in vitro studies. There is limited discussion of clinical trials or human cohort validations, which weakens the translational relevance of the proposed strategies.
 Integrate clinical evidence (e.g., biomarker studies in patient cohorts) to bridge mechanistic insights with therapeutic potential.
 The disease-specific sections (e.g., AKI, CKD, DKD) repetitively describe m6A’s role in inflammation, oxidative stress, and fibrosis without sufficiently differentiating mechanistic nuances across pathologies. Streamlining these sections or emphasizing disease-specific regulatory networks (e.g., METTL14 in DKD vs. METTL3 in RCC) could enhance clarity.
 Some important literature was missed, such as studies highlighted the m6A role in non-communicable diseases (PMID: 38839892).
 The review would benefit from incorporating recent literature on RBPs, particularly studies exploring the role of FXR1 as a new m6A reader protein: (PMID: 38238286, PMID: 39511680, https://doi.org/10.36922/gpd.5068).

Experimental design

no comment

Validity of the findings

no comment

Additional comments

no comment

Reviewer 2 ·

Basic reporting

In the manuscript entitled “The Role of N6-methyladenosine (m6A) RNA Methylation Modification in Kidney Diseases: From Mechanism to Therapeutic Potential”, the authors provide a comprehensive overview of the regulatory functions and therapeutic implications of m6A RNA methylation across various kidney diseases, including AKI, CKD, DN, RCC, and LN. They perform a structured literature review, highlighting key m6A-modifying enzymes (writers, erasers, readers) and examining their biological roles in both experimental and clinical settings.

While the manuscript discusses several signaling pathways, a more in-depth analysis, such as YAP and PTEN pathways in the context of kidney injury by referencing current studies would further strengthen the review. (PMID: 34515350, PMID: 39781453, PMID:34476213).

Additionally, the authors frequently rely on in vitro and animal model data without clearly addressing their translational relevance or highlighting the gaps in human-based validation. Clearly indicate the level of evidence for each disease section and differentiate preclinical from clinical data.

Experimental design

no comment

Validity of the findings

no comment

Additional comments

no comment

---

## Round 0.2 · accepted · Accept

The authors have addressed all of the reviewers' comments.

Reviewer 1 ·

Basic reporting

No further comments. The authors have satisfactorily addressed all my concerns and incorporated the suggested revisions.

Experimental design

-

Validity of the findings

-

Reviewer 2 ·

Basic reporting

The authors addressed my concerns.

Experimental design

-

Validity of the findings

-